# The Potential Influence of Group Membership on Outcomes in Indicated Cognitive-Behavioral Adolescent Depression Prevention

**DOI:** 10.3390/ijerph17186553

**Published:** 2020-09-09

**Authors:** Paul Rohde, Frédéric N. Brière, Eric Stice

**Affiliations:** 1Oregon Research Institute, Eugene, OR 97403, USA; 2École de Psychoéducation, Université de Montréal, Montreal, QC H3T 1J4, Canada; paulrl@ori.org; 3Psychiatry and Behavioral Sciences (Public Mental Health and Population Sciences), Stanford University Medical Center, Stanford, CA 94305, USA; estice@stanford.edu

**Keywords:** major depression, prevention, adolescents, group effects, clustering

## Abstract

Background: Adolescent depression prevention programs are typically delivered in groups in which adolescents share a common setting and interventionist, but the influence of the group is usually ignored or statistically controlled. We tested whether the primary outcomes of reductions in depressive symptoms and future onset of major depressive disorder (MDD) varied as a function of group membership. Methods: Data were available from two randomized trials in which 220 adolescents received the Blues Program indicated prevention intervention in 36 separate groups; participants were assessed at baseline, post intervention, and at 6-, 12-, and 24-month follow-ups. Results: Ten percent of participants had developed MDD 2 years post intervention. Group-level effects for MDD onset over follow-up were nonsignificant (accounted for <1% of variance; ICC = 0.004, ns). Group-level effects for depressive symptom change across the follow-up period were also nonsignificant (ICC = 0.001, ns) but group effects accounted for 16% of depressive symptom change immediately post intervention (ICC = 0.159, *p* < 0.05). Group-level clustering of posttest depressive symptoms was not associated with size of group or gender composition. Conclusions: Membership in specific adolescent cognitive-behavioral depression prevention groups may have an impact in terms of immediate symptom reduction but does not appear to have significant prevention effects in terms of long-term symptom change or MDD onset.

## 1. Introduction

Depression is a highly prevalent disorder that often first develops in adolescence [1] and is associated with significant global disease burden [2]. Several universal and targeted interventions aimed at preventing depression in young people have been developed and tested. Most have been manualized, group-based interventions based on cognitive-behavioral (CB) therapy, interpersonal psychotherapy (IPT), or third-wave CB content [3]. Recent reviews suggest that many depression prevention programs have been efficacious, with better results obtained in targeted relative to universal programs [3].

Because adolescent depression prevention is almost always delivered in groups (e.g., 88% of trials in [3]), adolescents who are placed in the same group are not independent. They share a common setting and interventionist, and are likely to influence each other. For these reasons, participants in depression prevention trials may have outcomes that are more similar to the other participants in their group compared to adolescents in other groups. This group-level phenomenon is known as a “clustering effect” and can be estimated using the intra-class correlation (ICC), which represents the ratio of group-level variation to total variation in the outcome [4]. Previous studies in adults have demonstrated the existence of significant clustering effects (or ICC) in group-based interventions (e.g., 6–7% of variance [5,6]).

Depression prevention research to date has focused on estimating intervention effects on individual-level outcomes. In this context, clustering effects need to be controlled when comparing intervention and control conditions because they create dependencies among observations that violate the independence of observations assumption of most statistical tests [7]. Approaches to remove the influence of clustering effects on individual-level analyses have been proposed and discussed in detail [7].

However, clustering effects also potentially have substantive clinical value. Indeed, the existence of significant clustering effects on key outcomes indicates that participants in some groups tend to do better, on average, than participants in other groups. The extent of clustering effects in adolescent depression prevention group interventions (and, to our knowledge, other adolescent prevention or treatment efforts) is unknown. To the degree that significant clustering exists, such group effects may be capturing important variance due to unique group processes that have been previously ignored but would warrant future attention.

The identification of group effects requires a specific statistical approach. Instead of controlling for clustering effects in individual-level analyses comparing experimental and control conditions, multilevel analyses can be used to examine whether group factors predict differential improvement within the intervention condition specifically [8]. Group factors may thus be conceptualized as group-level predictors of response to the intervention. Multilevel modeling allows investigators to simultaneously examine both group (L2) and individual (L1) factors that relate to the outcomes of interest.

In this study, we used multilevel modeling to examine primary outcomes of the Blues Program [9,10], with a focus on groups as the unit of analysis. The Blues Program is a brief indicated group-based CB intervention that targets adolescents with elevated depressive symptoms, and involves six 1-h group sessions. Participants learn to increase involvement in pleasant activities and reduce negative thoughts using cognitive restructuring. The program has reduced depressive symptoms and future major depressive disorder (MDD) onset over 2-year follow-up relative to an educational brochure control condition in a randomized controlled trial (RCT) [11] and produced significantly greater reductions in depressive symptoms compared to alternative credible interventions, including supportive expressive therapy and bibliotherapy [9]. It has also shown positive effects in a subsequent effectiveness RCT, in which the program was delivered by school staff, although effects were found at posttest for symptom reductions and at 6-month follow-up for reduced MDD onset [10] but not through 2-year follow-up [12]. In the present study, we combined data from these two RCTs to maximize the number of intervention groups.

Our aim was to examine clustering effects for change in depressive symptoms and onset of major depressive disorder (MDD) over a 2-year period, the two primary program outcomes. To do so, we derived intra-class correlations (ICCs) for each outcome after receipt of the program, adjusting for baseline depressive symptoms. This allowed us to estimate the relative importance of group factors versus individual factors in explaining why some participants experienced better outcomes than others.

## 2. Materials and Methods

### 2.1. Participants

We combined data from efficacy and effectiveness RCTs evaluating the Blues Program. Participants were randomized to the CB group intervention or alternative conditions. As noted previously [13], the two studies did not differ in rates of attrition, receipt of adjunctive mental health treatment over follow-up, or the impact of intervention condition on depressive symptom growth. In the present report, we included only the 220 adolescents from both trials who received the Blues Program (62% female, 60% European American, mean age of 15.5 years (*SD* = 1.2, range = 13–19). These participants were clustered in 36 groups of 3 to 10 participants (mean group size = 6.49; *SD* = 1.59). The two trials are briefly described next. More information on the study designs can be obtained in the primary outcome reports listed below.

### 2.2. Trial 1: High School Efficacy Trial

The high school efficacy trial (completed between 2004 and 2007) included 341 students from six Texas high schools who were targeted as a function of elevated depressive symptoms (scores greater than 20 on the Center for Epidemiologic Studies Depression Scale (CES-D; [14]) [9]. The sample was 56% female, 46% European American, and had a mean age of 15.6 years (*SD* = 1.2, range = 14–19). Participants were randomized to the Blues Program, a supportive expressive group, CB bibliotherapy (self-help book), or an educational brochure control. Facilitators were pairs of clinical psychology graduate students. Assessments were conducted at baseline, posttest (i.e., within one week after the last Blues Program session), and at 6-, 12-, and 24-month follow-ups. In the present study, we included the 89 participants who were randomized to a Blues Program group; these participants were clustered in 14 groups.

### 2.3. Trial 2: High School Effectiveness Trial

The high school effectiveness trials (completed between 2009 and 2011) included 378 students from five Oregon high schools who were recruited on the basis of a modified CES-D collected by school staff [10]. The sample was 68% female, 72% European American, and had a mean age of 15.5 years (*SD* = 1.2, range = 13–19). Participants were randomized to the Blues Program group, CB bibliotherapy, or educational brochure control. Facilitators were pairs of staff at each high school who were trained by the research team. Assessments were conducted at baseline, posttest, and at 6-, 12-, 18-, and 24-month follow-ups. In the present study, we included the 131 participants randomized to a Blues Program group; these participants were clustered in 22 groups.

### 2.4. Intervention

The structure and content of the Blues Program CB group intervention was identical in both trials, though the manual was more detailed for school staff in Trial 2. The six 1-h sessions were conducted at schools weekly in single-sex groups. Sessions focused on enhancing participants’ engagement in pleasant activities and identifying negative cognitions in order to replace them with more positive cognitions. Pairs of research clinicians (i.e., 10 graduate students in a mental health discipline) facilitated the 14 groups in Trial 1; pairs of 13 school counselors or nurses facilitated the 22 groups in Trial 2. If a participant missed a session, a brief make-up session was conducted. Intervention content and facilitator training are presented in detail elsewhere [9,10].

### 2.5. Outcomes

Depressive symptoms and DSM-IV-R [15] MDD diagnoses were assessed by research staff using 16 items from the semi-structured Schedule for Affective Disorder and Schizophrenia for School-Age Children (K-SADS; [16]). Assessors interviewed participants for the presence and severity of each MDD symptom current at baseline (defined as “in the last month”) and since the last assessment on a monthly basis at subsequent assessments. Items used a four-response format (1 = not at all to 4 = severe symptoms; with ratings of 3 and 4 reflecting diagnostic levels). Severity ratings for each symptom were averaged. The present study showed inter-rater reliability for depression symptom items (κ = 0.90) and diagnoses (κ = 0.93). Assessors, masked to condition, had at least a BA in psychology, received 40 h of training in interviewing, and demonstrated high inter-rater agreement (κ > 0.80) based on expert ratings of training interviews. They also had to demonstrate inter-rater κ values >0.80 for a randomly selected 10% of taped interviews throughout the study.

### 2.6. Statistical Analyses

We first examined group clustering effects in the cumulative onset of MDD over the 2-year follow-up. We fitted a two-level logistic model in Mplus 7.21, adjusting for baseline depressive symptoms and Trial (efficacy vs. effectiveness). We derived the ICC as a measure of group clustering (group variation over total variation) using the standard formula for logistic models [17]. We also examined MDD onset at specific follow-up points. To examine potential clustering effects of depressive symptoms, we first fitted a three-level model, with Time (Level 1) nested within Individuals (Level 2) nested within Groups (level 3), adjusting for baseline depressive symptoms and Trial. We then examined clustering effects at specific follow-up points using two-level models.

## 3. Results

### 3.1. Clustering Effects in Primary Outcome #1: Cumulative MDD Onset

A total of 22 participants (10%) had developed MDD 2-years post intervention (*n* = 9, 10% in Trial 1 and *n* = 13, 10% in Trial 2). As shown on the left side of Table 1, less than 1% in the variation of cumulative MDD onset was explained by groups, producing a nonsignificant ICC (ICC = 0.004, ns). We attempted to examine MDD onset at specific follow-up points, but these models did not converge because of low rates of disorder onset.

### 3.2. Clustering Effects in Primary Outcomes #2: Depressive Symptoms

We next examined group clustering effects in depressive symptoms (right side columns of Table 1). No variation in depressive symptoms was explained by Groups over the full follow-up period (ICC = 0.001, ns). When we examined clustering effects at specific follow-up points using two-level models, Groups explained 16% of the variation in depressive symptoms at posttest (ICC = 0.159, *p* < 0.05). No clustering effects were found for change in symptom levels from baseline to the subsequent follow-up points.

Regarding the nature of the change that occurred from baseline to posttest, average depressive symptom levels among Blues Program participants were 1.58 (*SD* = 0.35) at baseline and 1.45 (*SD* = 0.32) at posttest, suggesting a slight reduction in symptom levels. Conversely, functionally no change occurred from baseline to posttest for the brochure control participants, who were not included in this report (their mean pre- and post-scores were 1.55 (*SD* = 0.34) and 1.58 (*SD* = 0.39), respectively).

### 3.3. Exploratory Analyses

Given that significant group-level clustering occurred only for depressive symptoms immediately post intervention, we explored whether the number of adolescents in the group (group size) or gender composition predicted group-level clustering of posttest depressive symptoms, controlling for baseline symptoms. These two variables were selected because group size varied quite a bit (from 3 to 10 participants) and because groups were either all female or all male; we did not have data on common treatment process measures, such as therapeutic alliance or group cohesion. We tested two multilevel models, adding the level 2 predictor of interest (i.e., either group size or gender composition) to models tested for primary analyses. Neither group size (*p* = 0.57) nor gender composition (*p* = 0.86) was found to predict group-level clustering of posttest depressive symptoms.

## 4. Discussion

In this study, we conducted the first detailed examination of group effects in adolescent depression prevention research. We focused on 36 groups from two RCTs of the Blues Program indicated CB prevention group intervention [9,10]. We tested for the effects of specific group membership on the primary outcomes of depressive symptom change and onset of MDD.

Our results showed that group effects explained a significant proportion of posttest depressive symptom levels (16%) even after controlling for baseline symptoms. This group-level variation is higher than variation reported in previous adult treatment studies (6–7%) [5,6,19]. In other words, some groups appeared more successful than others at reducing existing depressive symptoms during receipt of the group intervention. This finding suggests that group-level factors do play a role in explaining the success of an intervention, although the majority of variation in outcomes is related to individual-level differences. As a comparison, approximately 5% of the variation in outcomes for adult patients across their course of treatment has been attributed to therapist effects [19]. Regarding the nature of the change, average depressive symptom levels for the Blues Program participants were somewhat lower at posttest compared to baseline, whereas depression symptom levels neither decreased nor increased notably among brochure control participants. Thus, this acute phase change appears to be a small “treatment effect” (i.e., depressive symptom reductions from baseline to post intervention) rather than a truly preventive effect, which might have been detected if the brochure control group participants had shown significant symptom escalation from baseline to posttest that was not seen for Blues Program participants.

No group-level variation could be found in depressive symptoms at later time points or in MDD onset. If detected, differences in MDD onset would have meant that group membership explained a portion of the true prevention effects of the program. Rather, the pattern of the present findings suggests that groups have a small effect that is detectable during participation but that they do not have a lasting, preventative effect. It has been argued that the achievement of prevention effects over follow-up is more meaningful than short-term treatment effects for youths with subthreshold symptoms (e.g., [20]). It is possible that it might be necessary to have a larger sample size of groups and subsequently greater statistical power to identify group effects on long-term outcomes, if they are small in magnitude.

Given the focused nature of the significant effects, we explored whether two group-level characteristics were correlated with group-level clustering. Neither number of participants in the group or gender composition had a significant effect. Future studies should assess group process measures occurring during the sessions (e.g., group cohesion, universality, installation of hope) that might predict intervention outcomes. If significant, knowledge of these processes might be used to modify the intervention or its implementation to produce more consistently successful immediate outcomes. In general, investigating multiple levels of analysis provides a more complete understanding of treatment effects in group therapy [8] and is recommended for use in the prevention field, as appropriate.

Our study is innovative in many regards, but has important limitations. First, we combined data from an efficacy trial that achieved longer lasting depressive outcomes relative to the educational brochure condition than were achieved in the effectiveness trial (though rates of MDD onset over 2 years for the two trials were identical). Potentially, the combination of prevention groups delivered by research clinicians versus real-world school staff provided additional variation between groups that would enhance detection of group-level effects. Second, although we benefited from the combination of two large RCTs, we still had a fairly small number of intervention groups. This limited our power to detect effect sizes that small in magnitude, although such effects might be of limited clinical significance. Third, groups were more homogeneous than would be the case in non-research-based settings; there were restrictions in group composition factors (e.g., minimum and maximum group size, age of participants, risk status) due participation in a research trial. Additionally, the sample was somewhat restricted in terms of race, ethnicity, and socioeconomic status due to the sources of recruitment. Fourth, group facilitators were highly trained and supervised in the efficacy trial and received relatively frequent and detailed supervision in the effectiveness trial. Both of these factors presumably increased adherence to the manual and potentially general therapeutic competence, reducing the amount of variation related to group membership. Fifth, clustering effects can be a reflection of either group-related clinical factors or simply nuisance factors (e.g., seasonal effects).

## 5. Conclusions

To our knowledge, this is the first study to examine the amount of variance in outcomes that is associated with group membership from adolescent depression prevention programs, and we are unaware of research on the group clustering effect for prevention programs in other areas of adolescent psychopathology. Our results suggest that delivering depression prevention in a group format does not simply represent a convenient, cost-effective way to provide cognitive-behavioral interventions to at-risk adolescents, but that groups matter clinically, at least in the short term. Indeed, one of the reasons universal depression prevention programs tend to be less effective might be due to the absence of group effects (i.e., they are more often delivered system-wide or individually rather than in small groups). Our study suggests that depression indicated prevention groups appear to have a small but meaningful effect in terms of immediate symptom reductions during receipt of the intervention. That effect is valuable to document and to build on clinically. However, in terms of the much more important goal of achieving future prevention effects, we found no indications that some groups have better (or poorer) long-term outcomes, though research with larger, more naturalistic samples should be pursued.

## Figures and Tables

**Table 1 ijerph-17-06553-t001:** Clustering effects.

	MDD Onset	Depressive Symptoms				
		Full Follow-Up	Posttest	6 Months	12 Months	24 Months
Group variation, τ00	0.001	0.001	0.013 *	0.005	0.009	0.001
Participant variation, σ^2^	-	0.028 **	0.070 ***	0.127 ***	0.127 ***	0.147 ***
Intra-class correlation	0.004	0.001	0.1589 *	0.040	0.065	0.008

Note: The first two rows provide the amount of variation (unstandardized) in major depressive disorder (MDD) onset or depressive symptom change associated with the group level (level-3) and participant level (level-2). Not shown is the level of variation in outcomes at the level of time (level-1), which was estimated at 0.098 and statistically significant at *p* < 0.001. Intra-class correlations were computed as τ_00/_(τ_00 +_ σ^2^), as is described in [18]. *** *p* < 0.001; ** *p* < 0.01; * *p* < 0.05.

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
