# Peer review of "The Potential Influence of Group Membership on Outcomes in Indicated Cognitive-Behavioral Adolescent Depression Prevention"

_ijerph, 2020, doi:10.3390/ijerph17186553_

Round 1
Reviewer 1 Report
This study tackles an important issue, i.e. the influence of group membership in the outcomes of a depression prevention program. The methods adopted are adequate and generally well presented. However, I have some concerns, listed below.
Materials and Methods
Participant information is presented in the 2.1, 2.2 and 2.3 paragraphs. However, while descriptive statistics for gender and age are presented for the whole samples of the two trials (paragraphs 2.2 and 2.3), no such information is provided for the subsample that was examined in this study. I suggest to include such data in paragraph 2.1.
Lines 127-131 read “Depressive symptoms and DSM-IV-R [15] MDD diagnoses were assessed by research staff using 16 items from the semi-structured Schedule for Affective Disorder and Schizophrenia for School-Age Children (K-SADS; [16]). Participants indicated the severity of each symptom over their lifetime (Trial 1) or the past 12 months (Trial 2) at baseline and since the last assessment on a monthly basis at subsequent assessments”. I suggest to explain more in detail the measures adopted and the relation between self report (“participants indicated…”) and assessment by research staff. Moreover, the fact that in Trial 1 participants were asked to rate the severity of symptoms over their lifetime, while in Trial 2 they were asked to refer to the last 12 months might introduce an heterogeneity that should be discussed.
Results
I suggest to report effect sizes and exact p-values within the main text, specifying whether the former are standardized or unstandardized. In Table 1 the note should include what the coeffients reported refer to. The same should be done in the abstract, reporting b or beta, as well as the exact p value (line 23).
Author Response
Dear reviewer, please find author's responses in the attachment.

Reviewer 2 Report
Thank you for giving me the opportunity to review the article. The authors conducted a study on the potential influence of group membership on outcomes in indicated cognitive-behavioral adolescent depression prevention. The topic may be socially important. However, the authors did not explain statistical methods used in this study in the manuscript. It is very important to clarify the methods especially this kind of investigation. For example, NO “statistical analysis” sub-section is in the Materials and Methods section. The authors did not mention any cut-off value of statistical significance and categories of ICC. Therefore, I thought that the manuscript cannot be accepted for publication in the journal IJERPH. I strongly recommend the authors to check the manuscript and add the process of this research in the manuscript in detail before submitting to other journals.
Author Response

(The authors gave the same response as above.)

Reviewer 3 Report
I sincerely congratulate the authors for the great work done. Studies like this one are needed to understand better how group interventions work. Not only for assessing efficacy or effectiveness, but more importantly, to help to find those little issues that could explain some conflicting results in group interventions. Moreover, the study finds that, overall, the Blues Program CB has significant individual effects, beyond the main group clustering effect found. I only have two little comments. I feel that the references are a bit old. Please, up to date the references. It is hard to defend nowadays that depression "often first develops in adolescence" (P1, L31) with a reference of 2005. Besides, please, add more references to some sentences. There is a great bulk of scientific literature to defend the usefulness of multilevel methods, for example. But, out of these comments, I repeat, congratulations.Author Response
Dear reviewer, please find author's responses in the attachment.
